# Novel Oxygen- and Curcumin-Laden Ionic Liquid@Silica Nanocapsules for Enhanced Antimicrobial Photodynamic Therapy

**DOI:** 10.3390/pharmaceutics15041080

**Published:** 2023-03-28

**Authors:** Joana Henriques, João Pina, Mara E. M. Braga, Ana M. A. Dias, Patrícia Coimbra, Hermínio C. de Sousa

**Affiliations:** 1Chemical Process Engineering and Forest Products Research Centre, Department of Chemical Engineering, University of Coimbra, 3030-790 Coimbra, Portugal; 2Coimbra Chemistry Centre-Institute of Molecular Sciences, Department of Chemistry, University of Coimbra, 3004-535 Coimbra, Portugal

**Keywords:** silica nanocapsules, curcumin, ionic liquids, oxygen storage/release, photodynamic inactivation

## Abstract

The efficiency of photodynamic therapy is often limited by the scarcity of oxygen at the target site. To address this problem, this work proposes the development of a new nanosystem for antimicrobial photodynamic therapy applications (aPDT) where the natural-origin photosensitizer curcumin (CUR) is immersed in an oxygen-rich environment. Inspired by the perfluorocarbon-based photosensitizer/O_2_ nanocarriers reported in the literature, we developed a new type of silica nanocapsule containing curcumin dissolved in three hydrophobic ionic liquids (ILs) with high oxygen dissolving capacities. The nanocapsules (CUR-IL@ncSi), prepared by an original oil-in-water microemulsion/sol-gel method, had a high IL content and exhibited clear capacities to dissolve and release significant amounts of oxygen, as demonstrated by deoxygenation/oxygenation studies. The ability of CUR-IL solutions and of CUR-IL@ncSi to generate singlet oxygen (^1^O_2_) upon irradiation was confirmed by the detection of ^1^O_2_ phosphorescence at 1275 nm. Furthermore, the enhanced capacities of oxygenated CUR-IL@ncSi suspensions to generate ^1^O_2_ upon irradiation with blue light were confirmed by an indirect spectrophotometric method. Finally, preliminary microbiological tests using CUR-IL@ncSi incorporated into gelatin films showed the occurrence of antimicrobial effects due to photodynamic inactivation, with their relative efficiencies depending on the specific IL in which curcumin was dissolved. Considering these results, CUR-IL@ncSi has the potential to be used in the future to develop biomedical products with enhanced oxygenation and aPDT capacities.

## 1. Introduction

Photodynamic therapy (PDT) is a biomedical therapeutic approach that combines three components that alone usually do not elicit a toxic response in biological systems: a photosensitive compound, a light source with a well-defined wavelength, and molecular oxygen (O_2_) [1,2]. PDT is based on the occurrence of a non-thermal photodynamic reaction in which, upon light irradiation with a suitable wavelength, the photosensitizer (PS) is activated, reacts with O_2_, and generates reactive oxygen species (ROS) (through Type I and Type II reactions) that induce a cytotoxic effect on the desired target cells [1,2,3]. Compared to other cytotoxic therapies, PDT has the major advantage of exhibiting double selectivity: PS can be designed to preferentially accumulate in specific target cells or tissues, and light can be applied exclusively to the specific regions to be treated and for pre-determined treatment periods [1,2]. This spatiotemporal selectivity makes PDT a therapy that has relatively few, non-severe side effects and that may be suitable for repeated applications in situations of recurrence or multiple lesions.

Despite its advantages, PDT has several shortcomings. The typical low chemical stability and hydrophobicity of most PSs are two examples of these limitations [3,4]. Since biological systems are aqueous environments, hydrophobic PSs are prone to aggregate and change their photochemical properties, thus reducing their ROS production efficiency [3,4]. To address these issues, the incorporation of PSs into stable and non-toxic nano-vehicles has been one of the most employed methodologies to enhance PS bioavailability and to improve PDT efficacy [2,3,5]. This general approach can also enable active cellular targeting, which is mostly achieved by chemical or physical surface functionalisation of nanoparticles (NPs) with specific chemical ligands [5,6,7]. Several recent reviews are available in the literature reporting the use of numerous types of NPs in PDT applications [3,4,5,8,9], such as inorganic NPs [4], polymeric NPs [8], liposomes [9,10], micelles [11,12], and metal-organic frameworks (MOFs) [7]. Another relevant aspect that may hinder PDT efficiency is oxygen scarcity in pathogenic tissues. To address this issue, PS-loaded NPs with oxygen storage or oxygen generating properties have already been proposed [13]. One example is PS-loaded nanocapsules incorporating perfluorocarbons, a class of fluorinated compounds with high oxygen solubility and capture/release capacities [10,11].

Although most applications of PDT are essentially focused on anticancer therapies, and therefore most PS-loaded NPs have been developed for these purposes, the photodynamic effect has also been explored for the inactivation of microorganisms [14,15], being referred to as antimicrobial photodynamic therapy (aPDT). In this context the use of PS-loaded NPs has also found relevant applications [16], particularly for infected wound management, where light-activated nanoparticles with photodynamic and/or photothermal properties have been combined with polymeric materials to produce antimicrobial multifunctional wound dressings [17,18]. Nevertheless, other biomedical products using this type of antimicrobial nanosystem can be easily envisaged, such as aqueous suspensions, hydrogels, creams/ointments, bioadhesives, bone cements, device coatings, etc.

Considering these potential applications, in this study we present light-responsive silica-based nanocapsules with the ability to generate oxygen singlet in the presence of molecular oxygen and upon irradiation with blue light. These nanosystems consist of a porous and light-transparent silica shell encapsulating different hydrophobic ionic liquids (ILs), in which curcumin, a natural-origin PS, is dissolved. In addition to their curcumin-solubilizing properties, the selected ILs have relatively high oxygen dissolving capacities [19,20,21], thus enabling the maintenance of high oxygen levels around the PS molecules, which is an indispensable factor for the effectiveness of PDT and aPDT.

Porous silica-based NPs, including silica-based nanocapsules (ncSi), have been extensively tested for the immobilization and/or encapsulation of bioactive substances, PSs, and imaging agents due to a well-known set of favourable properties for these applications such as low toxicity and good biocompatibility, chemical and mechanical stability, transparency, micro/mesoporosity, and a surface chemistry that allows functionalisation/bioconjugation through a series of chemical (and physical) modifications [22,23].

Curcumin is a naturally occurring PS that has been extensively studied for aPDT applications, primarily due to its inherent ability to generate ROS, particularly singlet oxygen, upon exposure to blue light [24]. Curcumin also exhibits several other important and well-known biological properties, such as anticancer and anti-inflammatory activities [25]. However, curcumin is a highly hydrophobic compound with a poor solubility and stability in aqueous solutions, which severely limits its straightforward use in most biological applications [25]. Furthermore, and due to self-aggregation in aqueous systems, curcumin in water usually exhibits a very low singlet oxygen sensitization quantum yield [26]. To improve its stability and bioavailability as well as its phototoxicity, curcumin has been encapsulated/immobilized in several NPs and other nanoformulations [6,25,27,28] using conventional processing oils/solvents and non-conventional hydrophobic solvents, such as deep eutectic solvents or ionic liquids [29,30].

Due to their chemical and physical properties, as well as their easy “tunability” features, it is known that a large number of ILs can be “designed” and synthesized in order to solubilize either some specific or a wide range of solutes. In addition to this, ILs may also have other favourable properties as processing solvents for a wide range of applications, including in the pharmaceutical and biomedical fields (such as their very low, or non-existent volatilities, or their potentially high chemical and thermal stabilities) [31,32]. Additionally, due to some of their specific chemical/physical properties, it is possible to find some particular ILs that can absorb significant amounts of different gases, including respiratory gases (such as CO_2_ and O_2_). In fact, several ILs (alone or in combination with other substances/materials) have been extensively studied and tested for a variety of different applications involving the capture, storage, transport, and/or release of gases, especially CO_2_ [19,33]. Some ILs containing (per)fluorinated anions (more common) or cations (less common) can exhibit particularly high O_2_ solubilization capacities and have been proposed as low volatility alternatives to the typical perfluorocarbons that have been studied for oxygen carrying applications in the medical field [19]. Therefore, herein we explored the use of three different hydrophobic ILs as hydrophobic solvents able to solubilize curcumin and to simultaneously provide local high O_2_ concentrations around curcumin. The selected ILs represent three different IL families/types—pyrrolidinium, imidazolium, and phosphonium, although all of them contain the same perfluorinated bis (trifluoromethylsulfonyl) imide anion: 1-butyl-1-methylpyrrolidinium bis(trifluoromethylsulfonyl)imide [BMPYRR][NTf_2_]; 1-octyl-3-methylimidazolium bis(trifluoromethylsulfonyl)imide [OMIM][NTf_2_]; and trihexyltetradecylphosphonium bis(trifluoromethylsulfonyl)imide [P_6,6,6,14_][NTf_2_].

In the last decade, the micro/nanoencapsulation of ILs in inorganic or polymeric porous/permeable shells has been proposed as a strategy to overcome some capture/release problems related to the kinetic limitations usually imposed by the typically poor mass transport properties of ILs in many practical applications [33]. So far, this strategy has been explored in the fields of chemical synthesis and biocatalysts, removal of some water pollutants, energy storage, and, in particular, for CO_2_ capture and separation applications. In contrast, the potential applications of encapsulated ILs in the pharmaceutical and biomedical fields remain practically unexplored [31,33]. In fact, and to the best of our knowledge, the encapsulation of liquid solutions containing a PS and an IL for PDT or aPDT applications has never been reported in the literature, which highlights the innovative nature of the strategies/materials reported in this work.

## 2. Materials and Methods

### 2.1. Materials

Ionic Liquids (ILs) 1-butyl-1-methylpyrrolidinium bis(trifluoromethylsulfonyl)imide ([BMPYRR][NTf_2_], 99% purity), 1-octyl-3-methylimidazolium bis(trifluoromethylsulfonyl)imide ([OMIM][NTf_2_], 99% purity), and trihexyltetradecylphosphonium bis(trifluoromethylsulfonyl)imide ([P_6,6,6,14_][NTf_2_], >98% purity) were acquired from IoLiTec GmBH, Heilbronn, Germany. Tetraethyl orthosilicate (TEOS, >99% purity), co-surfactant/co-precursor (1H, 1H, 2H, 2H-perfluorooctyl) triethoxysilane (TEOFSilane, 98% purity), polyethylene glycol sorbitan monooleate (Tween^®^80), 4-(2-hydroxyethyl)-1-piperazineethanesulfonic acid (HEPES, >99.5% purity), curcumin (extract from *Curcuma longa*, ~80% purity; the last 20% contain other curcuminoids), gelatin (from porcine skin, type A, bloom 300), and phosphate buffered saline tablets (PBS) were purchased from Sigma Aldrich, Barcelona, Spain. 1,3-diphenylisobenzofuran (DPBF, >95% purity) was purchase from TCI Europe N.V., Zwijndrecht, Belgium. *Lactobacillus rhamnosus* (DSMZ 20,021) was obtained from DSMZ (German Collection of Microorganisms and Cell Cultures GmbH, Braunschweig, Germany) and Man, Rogosa and Sharpe (MRS) broth was provided by Scharlab, Barcelona, Spain. All other used chemicals were of analytical grade.

### 2.2. Synthesis of CUR-IL@ncSi Nanocapsules

Curcumin (CUR) and IL-loaded silica nanocapsules were prepared by a combined oil-in-water microemulsion/sol-gel method using an original and dual-purpose co-surfactant/silica co-precursor agent.

Curcumin was dissolved in the tested ILs (2 mg of curcumin per gram of IL) at room temperature. Surfactants Tween^®^80 and TEOFSilane (in a 3:1 mol:mol relative composition and in a 10:1 mol:mol ratio concerning the amount of IL) were dissolved in Milli-Q water (48 mL). Then, 0.5 mL of each curcumin/IL solution was added to the surfactants solution and the mixture was emulsified for 10 min using an ultrasound processor (Sonics VCX750, 750 W, 20 kHz, Sonics & Materials Inc., Newtown, CT, USA) operating at 30% amplitude in a 1 s pulse on/off mode. Each CUR-IL/water emulsion obtained was acidified with 0.5 M HCl to pH ~2.1, to hydrolyse the co-surfactant/silica co-precursor TEOFSilane, and left under magnetic stirring for 45 min. At the same time, 0.3 mL of TEOS was added to 10 mL of HCl acidified Milli-Q water (pH ~2.1) and hydrolysed under vigorous magnetic stirring (500 rpm) for 30 min. Subsequently, the hydrolysed TEOS solution was added to the CUR-IL/water emulsion, followed by the addition of 8 mL of a 0.1 M HEPES buffer solution. Finally, the pH of the emulsion was measured and adjusted to pH 7.3–7.5 with 0.5 M NaOH solution. The formation of the silica shell around CUR-IL/water emulsion micelles took place for 24 h, at room temperature, and under magnetic stirring (800 rpm). The formed CUR-IL-loaded silica nanocapsules (CUR-IL@ncSi) were isolated by centrifugation (1126 rcf, 5 min), washed twice with distilled water, frozen, and freeze-dried. In addition, nanocapsules containing only the tested ILs (IL@ncSi) were prepared (as controls) using the same procedures described above. A schematic representation of this procedure is shown in Figure 1.

### 2.3. Nanocapsules Characterization

The silica and IL contents of the prepared CUR-IL@ncSi nanocapsules were estimated by thermogravimetric analysis with a simultaneous TGA/DSC analyzer (SDT) Q600 from TA Instruments (New Castle, DE, USA). In brief, 5–10 mg samples were heated from room temperature up to 600 °C, under a nitrogen flux, at a constant heating rate of 10 °C/min. The mass remaining at 600 °C was assumed to correspond to the amount of silica present in the nanocapsules. In addition, the relative amount of ILs in the prepared ncSi were also determined by elemental analysis (CHNS-O analyzer, model EA1108, Fisons Instruments, Glasgow, UK), by quantifying the elemental sulfur present in the samples due to the [NTf_2_] anion.

The particle sizes of the CUR-IL/water emulsions and of the resulting CUR-IL@ncSi nanocapsules were analyzed by Dynamic Light Scattering (DLS) in a Zetasizer Nano ZS (Malvern Instruments, Malvern, UK).

The morphology of the CUR-IL@ncSi nanocapsules was assessed microscopically by SEM/STEM (FE-SEM Zeiss Merlin Gemini 2, CarlZeiss AG, Oberkochen, Germany).

### 2.4. Oxygen Loading/Release Experiments

The oxygen loading/release capacity of CUR-IL@ncSi nanocapsules was assessed using a protocol and experimental setup similar to that described by Lambert et al. [34]. First, a glass vial containing 30 mL of deionized water, under magnetic stirring (100 rpm) and with an immersed oxygen probe (SZ10T Consort Dissolved Oxygen Electrode), was bubbled with nitrogen (2 bar) until the oxygen concentration in the water reached a constant value, close to zero. At the same time, a suspension of CUR-IL@ncSi nanocapsules (40 mL) was oxygenated by bubbling oxygen (2 bar) for 20 min under magnetic stirring (100 rpm). The concentration of nanocapsules in the suspension corresponded to an IL to water ratio of approximately 1:80 *v*/*v*. Immediately after closing the O_2_ flow, a sample of 10 mL of the CUR-IL@ncSi nanocapsule suspension was taken with the help of a syringe and injected into the vial containing the degassed water, which was immediately isolated from the surrounding atmosphere with Teflon sealing tape. The evolution of the oxygen concentration in the release medium over time was then followed and recorded using an oxygen probe connected to a Consort C3010—Multi-Parameter analyzer (Consort, Turnhout, Belgium). The temperature was kept constant (25 °C) during both loading/release experiments, by placing the vials in glass thermal jackets connected to a thermostatically controlled bath (model CC-K6, Huber, Offenburg, Germany). For comparison, the oxygen loading/release capacities of the corresponding CUR-IL/water emulsions (IL:water ~1:96 *v*/*v*) were also studied. Oxygenated deionized water was used as a control. All experiments were performed in triplicate using samples (nanocapsules and emulsions) prepared less than one week before.

### 2.5. Singlet Oxygen Generation Experiments

The curcumin capacity to generate singlet oxygen (^1^O_2_) when dissolved in the studied ILs was determined by the direct measurement of phosphorescence at 1275 nm, after sample irradiation with a 355 nm Nd:YAG laser. The experimental setup is described elsewhere [35]. The samples analyzed were: (i) air-saturated curcumin/IL solutions (1 mg/g_IL_); and (ii) air-saturated aqueous suspensions of CUR-IL@ncSi nanocapsules (~12 mg/mL). Singlet oxygen lifetimes were determined by fitting the experimental data (phosphorescence intensity at 1275 nm vs. time) with a first-order exponential model.

The capacity of CUR-IL@ncSi nanocapsules to release singlet oxygen into its aqueous surrounding when irradiated with blue light was determined by an indirect method using the chemical probe 1,3-diphenylisobenzofuran (DPBF) [36]. Samples were prepared immediately before use by mixing, in quartz cuvettes, 200 μL of DBPF fresh stock solution in ethanol (1 mM) with 1.8 mL of an aqueous CUR-IL@ncSi nanocapsule suspension (~6.6 mg/mL). The suspensions were kept under magnetic stirring in the dark for one minute. The cuvettes were then transferred to a 60 mm integrating sphere (Jasco, model ISV-722) in a JASCO 650 UV-VIS spectrophotometer (JASCO Corporation, Tokyo, Japan) and their initial absorbance spectra were recorded. The samples were then irradiated with a blue light (450 nm blue light LED Lamp, 5 W) placed at a distance of 82 cm, and their spectra were recorded after each 10 s of light irradiation. Spectra were also obtained for samples subjected to the same procedures but without blue light exposure (controls). In addition, and following the same procedures, the spectra of oxygenated samples (i.e., nanocapsule suspensions previously bubbled with pure O_2_ for 15 min) were recorded (in this case, the cuvettes were adequately sealed with airtight stoppers immediately after DPBF addition). Finally, for comparison purposes, and again following the same experimental procedures, the spectra of DPBF in the presence of solubilized curcumin (90:10 (*v*/*v*) water:ethanol) were also recorded.

The experiments were performed in triplicate for each type of sample tested. For each run, the absorbance intensities of DPBF at 416 nm were normalized with the absorbance of the non-irradiated sample (At/A_0_, where A_0_ is the absorbance of the non-irradiated sample) and plotted against irradiation time.

### 2.6. Antimicrobial Photodynamic Activity

The antimicrobial activity of the prepared nanocapsules was evaluated for freeze-dried CUR-IL@ncSi nanocapsules immobilized in a gelatin film. In brief, gelatin was dissolved in sterilized PBS (0.01 M, pH 7.4, 0.0027 M KCl, 0.137 M NaCl) at a concentration of 10% *w*/*v*, at 50 °C. Nanocapsules were then added to the gelatin solution (at the amount of 20% of the gelatin mass, 20 mg of CUR-IL@ncSi per mL of gelatin solution), and the resulting suspension was vortex-homogenized, cast in plastic Petri dishes (5 mL of the resulting viscous suspension), and left to dry overnight in a ventilated oven at 25 °C. The formed films (~150 µm thick) were cut into 1 cm diameter disks and laid at the bottom of the wells of a 48-wells cell culture plate. In addition to gelatin films containing CUR-IL@ncSi nanocapsules, other films were prepared by the same procedures to be used as controls: (i) films containing only gelatin; and (ii) films containing IL@ncSi nanocapsules (without curcumin).

*Lactobacillus rhamnosus* (*L. rhamnosus*) was used as a model for Gram-positive bacteria. *L. rhamnosus* were inoculated into 40 mL of MRS culture medium and grown overnight at 35 °C in an orbital shaker incubator. The resulting culture with a cell concentration of ~10^8^ CFU/mL was isolated, washed with PBS, and diluted with PBS to obtain a suspension with a cellular concentration of ~10^5^ CFU/mL. Then, 1 mL of this suspension was added to each well of the 48-wells cell culture plate already containing the gelatin discs to be tested. Wells without gelatin discs were used as controls. The plate was left to rest in the dark for 30 min and then irradiated with a blue light (450 nm blue light LED Lamp, 5 W), placed 20 cm above, for up to 2 h. At each irradiation time point (0, 1, and 2 h), 10 µL of cell suspensions were removed from each well and added to the wells of a 96-wells cell culture plate. After, 90 µL of MRS broth was added to each well and the 96-wells plate was placed in a Synergy HTX multi-mode microplate reader (BioTeck, Winooski, VT, USA), at 35 °C, where the absorbance of the samples at 600 nm was measured automatically over 24 h. Experiments were performed in triplicate for all systems/controls tested.

### 2.7. Statistical Analysis

Data are expressed as mean ± standard deviation. Comparisons between multiple groups were made with an analysis of variance (ANOVA) followed by a post-hoc Tukey test. Comparisons between two groups were made using a Student’s *t*-test. A *p*-value of <0.05 was considered to be statistically significant.

## 3. Results and Discussion

This work aims to develop novel, more efficient, and stable silica-shell nanocapsules that can be foreseen as light-responsive nanosystems for aPDT in biomedical applications. These nanocapsules contain a natural photosensitizer (curcumin) in their core, which is dissolved in hydrophobic ILs with relatively high oxygen dissolving capacities. Their liquid cores, together with their transparent and porous silica shells (which provide stability, transparency, and good oxygen/ROS diffusivity) offer essential advantageous conditions for the occurrence of more efficient photodynamic reactions when exposed to light.

The three hydrophobic ILs tested ([BMPYRR][NTf_2_], [OMIM][NTf_2_], and [P_6,6,6,14_][NTf_2_], represented in Table 1), were selected from a larger group of commercially available ILs, with the main criteria being their high water immiscibility (high hydrophobicity) and their high O_2_ solubility capacities (supported by the literature data). In addition, three of the most relevant IL families are represented in this selection: imidazolium-, phosphonium-, and pyrrolidinium-types of ILs. All these three cations, especially [P_6,6,6,14_] and [OMIM], have long alkyl chains, which are known to promote high hydrophobicity [37,38,39] and O_2_ solubility [20,21]. Furthermore, all these ILs contain the perfluorinated anion [NTf_2_], which is reported to be one of the most hydrophobic and O_2_-phylic anions [37,38]. An additional criterion for the selection of these ILs was their ability to dissolve relatively high amounts of curcumin (it was observed that curcumin was readily soluble in the three selected ILs and remained stable and active in solution for at least three months, when protected from light).

Nanocapsules were prepared using a combined oil-in-water microemulsion and sol-gel method developed and optimized by our research group and schematized in Figure 1. This method involves the emulsification of a CUR-IL solution in a continuous water phase containing a surfactant (Tween^®^80) and a co-surfactant/co-precursor (TEOFSilane). The latter has a highly hydrophobic perfluorinated C6 tail and a triethoxysilane head which, after hydrolysis, will play two different roles: as a co-surfactant, to stabilize CUR/IL micelles; and as a silica co-precursor, acting as an anchor point for condensation of water-soluble hydrolyzed TEOS around the CUR/IL emulsion droplets, thus leading to mechanically stable capsules. As such, a porous silica shell begins to form in the CUR-IL/water interface and the emulsion is converted to a nanocapsule suspension. The hydrolysis/condensation rates of the silica co-precursors (TEOS and TEOFSilane), as well as the resulting shell thickness/porosity and nanocapsules agglomeration/aggregation behavior are essentially controlled by the addition of the HEPES solution and by pH manipulation of the aqueous continuous phase of the CUR-IL/water emulsion. In this work, other experimental variables that can also affect the properties of the nanocapsules (reaction time, temperature, relative compositions (Tween^®^80:TEOFSilane, water, TEOS), stirring conditions, etc.) were maintained in all the experiments. On the other hand, the nanocapsules’ size and size distributions depend mainly on the initial emulsion droplet size (and droplet size distributions), which ultimately result from the chemical properties of the substances involved (IL, water, Tween^®^80, TEOFSilane), from their relative proportions, and from the emulsification conditions employed (e.g., temperature; emulsification method and specific operational conditions; energy provided vs. emulsification time, etc.).

### 3.1. Nanocapsules Characterization

The morphology of CUR-[BMPYRR][NTf_2_]@ncSi nanocapsules can be observed in Figure 2. SEM images show submicron particles with a broad size range. Fragments/empty capsules of larger size can also be observed, probably due to disintegration during their synthesis/processing or sample manipulation. Weiss et al. (2014) observed similar features in IL-containing silica capsules, also prepared by an oil-in-water emulsion/sol-gel method [43]. DLS results confirmed these obsservations, revealing a size distribution with two main broad size ranges (~100–200 nm, and ~600–1100 nm, Appendix A). Similar size distributions were found for CUR-[OMIM][NTf_2_]@ncSi and CUR-[P_6,6,6,14_][NTf_2_]@ncSi, i.e., broader distributions, with peaks centered around 60–100 nm and 350–550 nm, for CUR-[OMIM][NTf_2_]@ncSi (Appendix A), and 200–600 nm, for CUR-[P_6,6,6,14_][NTf_2_]@ncSi (Appendix A). These features are in good agreement with the results obtained for the corresponding emulsion droplet size distributions, which presented broad mono- or bimodal droplet size distributions, with average hydrodynamic sizes between 15–70 nm and 280–350 nm (see Appendix A).

STEM analysis (Figure 3) did not allow us to clearly prove the capsular features of the prepared nanocapsules, but the obtained STEM micrographs have confirmed their particle size ranges, namely for the smaller particle sizes in bimodal distributions.

The relative composition of the prepared nanocapsules was estimated by elemental analysis (EA) and thermogravimetric analysis (TGA) (Table 2). EA results (which quantify only the incorporated amounts of ILs) show that CUR-[OMIM][NTf_2_]@ncSi and CUR-[P_6,6,6,14_][NTf_2_]@ncSi nanocapsules have a very high IL content, about 83.6% and 94.4% *w*/*w*, respectively, while CUR-[BMPYRR][NTf_2_]@ncSi has a substantially lower IL content (35.7% *w*/*w*). TGA results follow the same general trend although different relative composition values were obtained since the TGA method (gravimetric) does not consider the specific relative amounts of the different organic species present in the samples (i.e., ILs, TEOFSilane, Tween^®^80, and CUR).

The different IL contents can be explained by the corresponding ILs’ solubilities in water ([BMPYRR][NTf_2_] > [OMIM][NTf_2_] > [P_6,6,6,14_][NTf_2_]), which determine the propensity of IL to leach into to the water continuous phase during nanocapsules synthesis, processing, and storage in aqueous media. In fact, the solubility of [BMPYRR][NTf_2_] in water is one order of magnitude higher than that of [OMIM][NTf_2_] (see Table 1). As for [P_6,6,6,14_][NTf_2_], although its experimental solubility in water is not available in the literature, it is expected to be the lowest of the three ILs, considering the high hydrophobicity of the [P_6,6,6,14_] cation (derived from its long alkyl chains) and the very low solubility of water in [P_6,6,6,14_][NTf_2_], as reported by Freire et al. (2008) [39].

### 3.2. Nanocapsules Oxygen Loading and Release Capacity

The capacity of the nanocapsules to load and release oxygen was evaluated by analyzing the evolution of the oxygen concentration in deoxygenated water after the addition of (i) an oxygenated CUR-IL@ncSi suspension sample; (ii) an oxygenated IL/water emulsion sample; and (iii) an oxygenated water sample (control). As can be seen in Figure 4, the moment each oxygenated sample is introduced the O_2_ concentration begins to rise, reaches a maximum, and then declines, converging to a plateau (~3–4 ppm). Since the experiment is performed in a closed flask, where the headspace above the liquid is also initially deoxygenated, the oxygen that is released and dissolved in the water phase will diffuse into the headspace above, until the O_2_ concentration in the liquid and gas phases reache equilibrium. This explains the descending segments of the curves. For the water control, the maximum O_2_ concentration (5.4 ppm) was reached after 2 min and 30 s. This time interval can essentially be attributed to the response time of the oxygen sensor since the homogenization and dilution of the two water fractions should be almost instantaneous.

The maximum O_2_ concentrations achieved for the studied samples ranged from 7.7 to 8.4 ppm, for nanocapsule suspensions, and 6.7 to 7.6 ppm, for IL/water emulsions. The time to reach the maximum O_2_ concentration ranged between 2 and 3 min, for nanocapsule suspensions, and 3 and 5 min for IL/water emulsions. These results clearly show that the maximum concentrations of dissolved O_2_ that were observed for the nanocapsule suspensions and the CUR-IL/water emulsions are always higher than the maximum concentrations obtained for the oxygenated water (control), which proves the capacities of these ILs to store and, subsequently, to release O_2_ to the water phase, either encapsulated in silica nanocapsules or as the disperse phase of IL/water emulsions. This reflects the higher capacity of ILs to dissolve oxygen in comparison to water (about 5 to 7 times higher, considering the values presented in Table 1 and the solubility of oxygen in pure water (1.22 mM, at 25 °C and 1 atm of O_2_ [44])).

For the same IL, and compared to CUR-IL/water emulsions, nanocapsule suspensions always lead to a higher maximum pO_2_ value and a shorter time to reach this value (Appendix A). This was not expected since the presence of a silica shell in nanocapsules should impose an additional barrier to mass transfer. Therefore, these results are probably due to the different IL:water (*v*/*v*) ratios that were employed for CUR-IL@ncSi (1:80) and for CUR-IL/water emulsions (1:96), which result in the dissolution of higher total amounts of O_2_ in nanocapsule suspensions than in emulsions, thus leading to enhanced initial O_2_ mass transfer gradients in the former. Indeed, for the same IL, the calculated overall volumetric mass transfer coefficients (K_L_a) for CUR-IL@ncSi were always higher than the corresponding coefficients for the CUR-IL/water emulsions (Appendix A), indicating that O_2_ transfer rates are favored when using nanocapsules. This is true even for the case of CUR-[BMPYRR][NTf_2_]@ncSi nanocapsules which have a lower IL relative composition (as concluded from TGA and EA results). The obtained K_L_a for CUR-IL@ncSi followed the trend: [OMIM][NTf_2_] > [P_6,6,6,14_][NTf_2_] > [BMPYRR][NTf_2_]. It was expected that these values would follow the O_2_ solubility trend for these ILs (Table 1); however, as already discussed, the relative IL amounts incorporated into the nanocapsules are quite different for the three studied systems, namely in the case of CUR-[BMPYRR][NTf_2_]@ncSi. The same justification (together with the observed standard deviation values) can explain the trend observed for K_L_a values of CUR-IL/water emulsions: [OMIM][NTf_2_] ≈ [P_6,6,6,14_][NTf_2_] > [BMPYRR][NTf_2_].

### 3.3. Singlet Oxygen Generation

The capacity of curcumin (CUR) to generate ^1^O_2_ when irradiated with appropriate light was confirmed by the direct detection of the ^1^O_2_ phosphorescence emission decay, collected at 1275 nm, using samples of CUR dissolved in bulk ILs (CUR-[BMPYRR][NTf_2_], CUR-[OMIM][NTf_2_], and CUR-[P_6,6,6,14_][NTf_2_]) or encapsulated into CUR-IL@ncSi nanocapsules. The time-resolved data (Figure 5) were fitted to an exponential model and the ^1^O_2_ lifetimes were calculated accordingly. The results are shown in Table 3. The ^1^O_2_ lifetime in solution strongly depends on its surroundings, namely the solvent [15]. For example, the ^1^O_2_ lifetime in water is about 3–4 µs while in deuterated water it is 57 µs [15]. The lifetimes of ^1^O_2_ in the bulk ILs were found to be much higher than in water: 35 µs ([BMPYRR][NTf_2_]), 28 µs ([P_6,6,6,14_][NTf_2_]), and 27 µs ([OMIM][NTf_2_]). In the literature, there are only a few reports concerning the generation of ^1^O_2_ by PS dissolved in ILs, but two of the already studied ILs are [BMPYRR][NTf_2_] and [OMIM][NTf_2_] [45,46]. The values reported in these works were 44.4 µs, for ^1^O_2_ generated by rose Bengal dissolved in [BMPYRR][NTf_2_] [45], and 34.5 µs, for ^1^O_2_ generated by methylene blue dissolved in [OMIM][NTf_2_] [46].

It was observed that the lifetimes of the ^1^O_2_ generated in the CUR-IL@ncSi nanocapsule suspension were considerably shorter than the lifetimes of ^1^O_2_ in the CUR-IL bulk solutions, reflecting the different environments probed by ^1^O_2_ during its lifetime. Indeed, depending on where ^1^O_2_ is generated (in the IL core, in the silica shells, or on the surface of CUR-IL@ncSi nanocapsules), these oxygen species will encounter different environments on their diffusion path before decaying, which will be reflected in its phosphorescence emission decay kinetics [47]. The similarity of the ^1^O_2_ lifetime generated in the [BMPYRR][NTf_2_]@ncSi nanocapsule suspension (4.5 µs) with the lifetime of ^1^O_2_ in water suggests that the medium in which the ^1^O_2_ phosphorescence decays is mainly an aqueous environment implying that ^1^O_2_ is essentially photosensitized in the nanocapsules shells, near to the surface, where it can rapidly diffuse into the aqueous surroundings. This implies that in CUR-[BMPYRR][NTf_2_]@ncSi nanocapsules, CUR is mainly trapped/adsorbed in the silica shells or in IL regions near the silica shell, rather than only in the IL bulk. This means that, despite the very low water solubility of CUR (≈5.4 ×10^−10^ mol/mol [30]), the relatively higher water solubility of [BMPYRR][NTf_2_] allows CUR to migrate from the core into the silica shell and water, being partially adsorbed on the silica shell by hydrogen bonding and electrostatic interactions between its OH groups and the silanol groups of silica [48]. On the contrary, due to the higher hydrophobicity of the other two ILs, one should expect more favorable interactions between these ILs and the highly hydrophobic CUR molecules. This may lead to their improved ability to retain CUR in their bulk and thus to the significantly higher ^1^O_2_ lifetimes observed for the CUR-[OMIM][NTf_2_]@ncSi and CUR-[P_6,6,6,14_][NTf_2_]@ncSi suspensions (12 µs and 16 µs, respectively). These differences may have some impact on the therapeutic performance of these nanosystems, as ^1^O_2_ must be released into its surroundings before decaying to the ground state to effectively react with the biological targets. The relationship between the distribution of photosensitizer molecules within a nanoparticle and its ability to release singlet oxygen has been demonstrated by Kabanov et al. (2018) [49]. The authors immobilized three PSs, with distinct degrees of hydrophobicity, in different regions of the mesoporous silica nanoparticles and demonstrated that the nanosystem having the PS distributed mainly on the surface released the major part of the produced ^1^O_2_ to its surroundings. On the contrary, the nanosystem in which the PS molecules were distributed throughout the depth of the nanoparticles released only a tiny amount of the produced ^1^O_2_ [49]. 

The capacity of CUR-IL@ncSi nanocapsules to release ^1^O_2_ into their aqueous vicinity upon irradiation with blue light was probed by an indirect chemical method, using 1,3-diphenylisobenzofuran (DPBF). DPBF reacts irreversibly with singlet oxygen causing a decrease in the intensity of its absorption spectrum around 420 nm. Since DPBF and curcumin have absorption bands in the same region (400–450 nm), the overlap of the two bands makes it more difficult to analyze the results (Figure 6a). In addition, the wavelength of the blue light used (~450 nm), which is close to the absorption maxima of DPBF, can also promote photodegradation. To investigate this effect, DPBF was dissolved in a mixture of EtOH:water (1:9 *v*/*v*). As shown in Figure 6b, the normalized absorbance at 416 nm of the aerated DPBF solution (with oxygen dissolved at atmospheric pressure) decays faster in the sequentially irradiated solution than in the solution kept in the dark, confirming the occurrence of photobleaching due to irradiation with blue light. However, even for the DPBF solution kept in the dark, it was observed that the absorbance intensity decreased which indicates the high photosensitivity of this molecule. Similar results were observed after irradiation of the DPBF + CUR solution (Figure 6b), indicating that the degradation of DPBF can be attributed mainly to the photobleaching effect rather than the photodynamic one, possibly due to the low ^1^O_2_ quantum photosensitization efficiency of curcumin in aerated aqueous solutions (about 0.01, according to Chignell et al. [26]).

Figure 6c shows the absorbance spectra of DPBF dissolved in CUR-[BMPYRR][NTf_2_]@ncSi suspensions (at equilibrium with air) upon successive irradiation with blue light. The normalized absorbance at 416 nm is plotted in Figure 6d, for irradiated and non-irradiated samples. It can be observed that, like the solution samples, the decrease in absorbance of the irradiated and non-irradiated nanocapsule suspensions is identical, indicating that, under the investigated conditions, the photobleaching of DPBF is the dominant effect rather than the reaction of DPBF with curcumin photosensitized ^1^O_2_. Identical results were obtained for CUR-[OMIM][NTf_2_]@ncSi and CUR-[P_6,6,6,14_][NTf_2_]@ncSi nanocapsule suspensions (Figure 6e,f, respectively).

On the contrary, for oxygen-saturated suspensions (blue plots in Figure 6d–f), a significantly faster absorbance decrease with irradiation time was observed for all three systems. This result can be attributed to the enhancement of ^1^O_2_ photosensitization by the curcumin incorporated in CUR-IL@ncSi nanocapsules due to the increase of oxygen concentration in the vicinity of curcumin molecules. Moreover, it also indicates that the generated ^1^O_2_ can diffuse to some extent from the nanocapsules to the aqueous surroundings, since the DPBF is in the aqueous phase.

### 3.4. Antimicrobial Photodynamic Effect

The potential antimicrobial photodynamic effect of CUR-IL@ncSi nanocapsules was tested against *L. rhamnosus*, a Gram-positive bacterium, with nanocapsules incorporated into gelatin films (see Appendix A for film photos and micrographs). Films containing CUR-IL@ncSi or IL@ncSi nanocapsules were immersed in bacterial suspensions and irradiated with blue light. A blank control (bacteria only) and a gelatin film (without nanocapsules) were used as controls. After contact and irradiation, bacteria suspensions were inoculated into a culture medium, and their growth was followed over time by measuring the optical density of the cultures at 600 nm [50].

No relevant statistical differences (*p* > 0.05) were observed in bacterial growth after bacteria were incubated with the different films for 30 min in the dark (Figure 7a), indicating that gelatin and nanocapsules per se do not have an acute harmful effect. Similarly, no significant antibacterial effect (*p* > 0.05) was observed in the control group (bacteria only) after exposure to blue light for one or two hours (Figure 7b).

The antibacterial photodynamic effect of the CUR-IL@ncSi nanocapsules was evaluated against IL@ncSi nanocapsules (without CUR) for different irradiation times. As expected, without irradiation (Figure 7c,f,i), no significant statistical differences (*p* > 0.05) were observed between the two groups at different incubation times. After 1 h of irradiation (Figure 7d,g,j), only the CUR-[P_6,6,6,14_][NTf_2_]@ncSi-loaded film showed significant differences in the OD_600_ at 10 and 12 h of incubation, which can be attributed to the photodynamic effect. After two hours of irradiation (Figure 7e,h,k), both CUR-[P_6,6,6,14_][NTf_2_]@ncSi- and CUR-[BMPYRR][NTf_2_]@ncSi-loaded films showed significant differences in bacterial growth. These differences were more pronounced for the bacteria exposed to CUR-[BMPYRR][NTf_2_]@ncSi, which may be a direct consequence of the relatively higher capacity of these nanocapsules to release ^1^O_2_, as proposed and discussed previously. As for CUR-[OMIM][NTf_2_]@ncSi-loaded films, no differences in bacterial growth were observed that could be attributed to the photodynamic effect, for 1 h and 2 h irradiation (*p* > 0.05). However, it is expected that these nanoparticles could exert the photodynamic antimicrobial effect under different experimental conditions, since their ability to generate ^1^O_2_ upon irradiation has been demonstrated. It should be noted that despite the inhibition of the growth rate, the bacteria still have the capacity to grow, which means that the photodynamic effect caused some damage but did not inactivate the total amount of bacteria.

This test has demonstrated the aPDT effect of CUR-[BMPYRR][NTf_2_]@ncSi and CUR-[P_6,6,6,14_][NTf_2_]@ncSi nanoparticles when incorporated into a polymeric support matrix. These results suggest that these nanoparticles can be used to develop biomedical products with antimicrobial capacities (due to their aPDT properties), such as wound dressings for wound infection management [17,18,50]. However, this was only a simple proof-of-concept and a qualitative test, which was performed with non-pathogenic bacteria. Quantitative colony counting tests with model Gram-positive and Gram-negative pathogenic bacteria will be performed in the future.

## 4. Conclusions

Three hydrophobic ILs with dissolved curcumin were successfully encapsulated in silica nanocapsules using a combined original IL-in-water microemulsion/sol-gel method. The ability of the nanocapsules to generate ^1^O_2_ upon appropriate irradiation was demonstrated by the directed detection of ^1^O_2_ phosphorescence at 1275 nm. The high oxygen solubility capacities of the encapsulated ILs gave the nanocapsules the ability to dissolve, retain, and release significant amounts of oxygen, as verified by oxygenation/deoxygenation studies. This creates an oxygen-rich environment around CUR molecules that favors the enhancement of ^1^O_2_ production, as demonstrated by the indirect spectrophotometric method using DPBF as a probe.

Preliminary microbiological tests with CUR-IL@ncSi nanocapsules incorporated into gelatin films showed the occurrence of an antimicrobial effect for CUR-[BMPYRR][NTf_2_]@ncSi- and CUR-[P_6,6,6,14_][NTf_2_]@ncSi-loaded gelatin films due to photodynamic inactivation.

Furthermore, the ^1^O_2_ lifetime for [BMPYRR][NTf_2_]@ncSi nanocapsules was the lowest and identical to the ^1^O_2_ lifetime in water (~4.0 µs), suggesting that in this sample ^1^O_2_ should be mostly photosensitized near the surface of silica shells, and thus CUR should be preferentially distributed on the [BMPYRR][NTf_2_]@ncSi nanocapsules silica shells. The relatively higher water solubility of [BMPYRR][NTf_2_], resulting in the lowest IL content due to partial leaching, could be responsible for the preferential distribution of CUR at the nanocapsules silica shell. Nevertheless, the distribution of curcumin inside the nanocapsules and, consequently, the site of singlet oxygen production and its ability to diffuse to the outside before decaying should be the subject of further studies as this aspect is not clearly elucidated. Future work should also include standard colony courting tests with model Gram-positive and Gram-negative pathogenic bacteria.

This work proves the potential of this new type of nanocapsule for aPDT applications and opens doors for future research, where different photosensitizers and ILs can be tested and the resulting IL@ncSi can be combined with other biomaterials to develop biomedical devices with enhanced oxygenation capacity and aPDT activity.

## Figures and Tables

**Figure 1 pharmaceutics-15-01080-f001:**
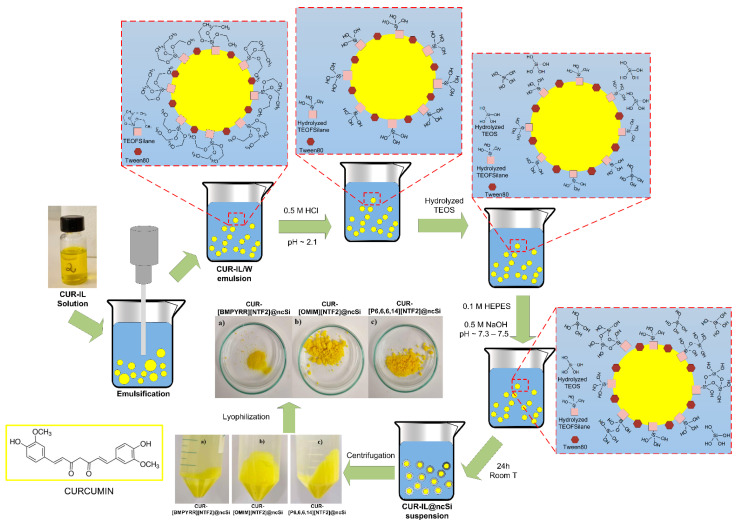
Schematic representation of the nanocapsule preparation process. Inserted pictures: (**a**) CUR-[BMPYRR][NTf_2_]@ncSi; (**b**) CUR-[OMIM][NTf_2_]@ncSi; and (**c**) CUR-[P_6,6,6,14_][NTf_2_]@ncSi.

**Figure 2 pharmaceutics-15-01080-f002:**
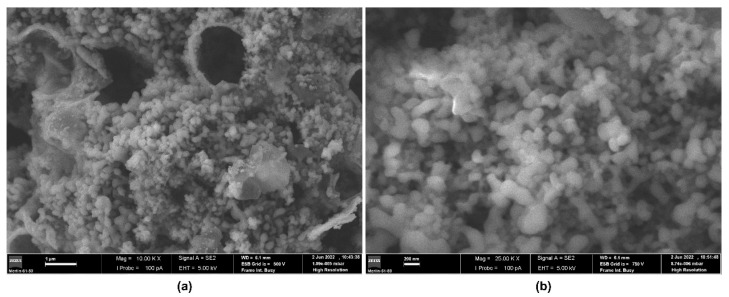
SEM images of freeze-dried CUR-[BMPYRR][NTf_2_]@ncSi. Scale Bars: (**a**) 1 µm; and (**b**) 200 nm.

**Figure 3 pharmaceutics-15-01080-f003:**
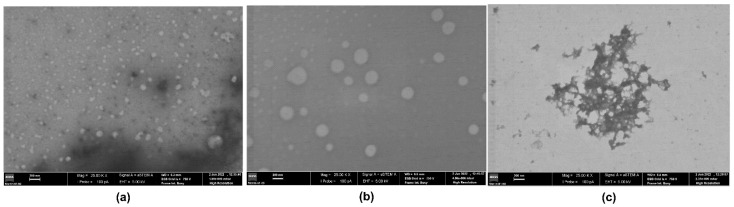
STEM images of: (**a**) CUR-[BMPYRR][NTf_2_]@ncSi; (**b**) CUR-[OMIM][NTf_2_]@ncSi; and (**c**) CUR-[P_6,6,6,14_][NTf_2_]@ncSi. Scale bar: 200 nm.

**Figure 4 pharmaceutics-15-01080-f004:**
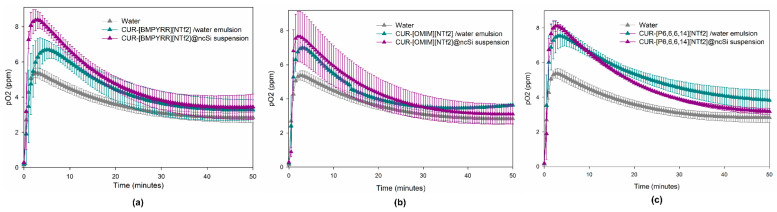
Evolution of the oxygen concentration in deoxygenated water after the addition of: (i) oxygenated CUR-IL@ncSi suspension samples; (ii) oxygenated IL/water emulsion samples; and (iii) oxygenated water samples (control). (**a**) [BMPYRR][NTf_2_]; (**b**) [OMIM][NTf_2_]; and (**c**) [P_6,6,6,14_][NTf_2_].

**Figure 5 pharmaceutics-15-01080-f005:**
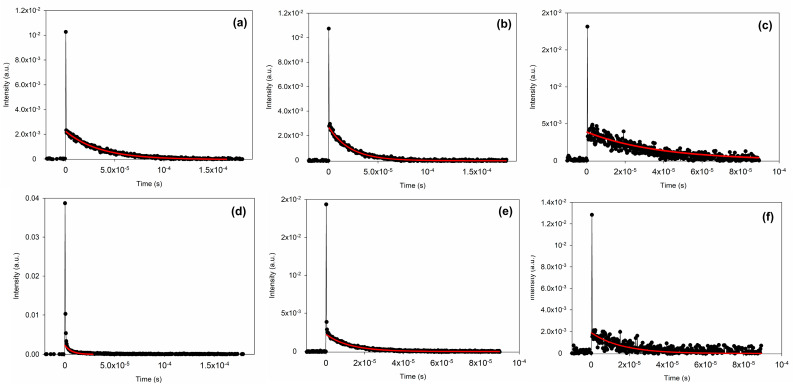
Singlet oxygen phosphorescence emission decays collected at 1275 nm after photosensitization with curcumin: (**a**) CUR-[BMPYRR][NTf_2_]; (**b**) CUR-[OMIM][NTf_2_]; (**c**) CUR-[P_6,6,6,14_][NTf_2_]; (**d**) CUR-[BMPYRR][NTf_2_]@ncSi; (**e**) CUR-[OMIM][NTf_2_]@ncSi; and (**f**) CUR-[P_6,6,6,14_][NTf2]@ncSi. Note: Due to the highly scattering medium of the solutions/suspensions and although a Newport long band-pass RG1000 filter was used, the initial part of the decays presented a spike that is attributed to scatter light. This feature was not considered in the kinetic analysis of the decays, which were well-fitted with a single-exponential decay law.

**Figure 6 pharmaceutics-15-01080-f006:**
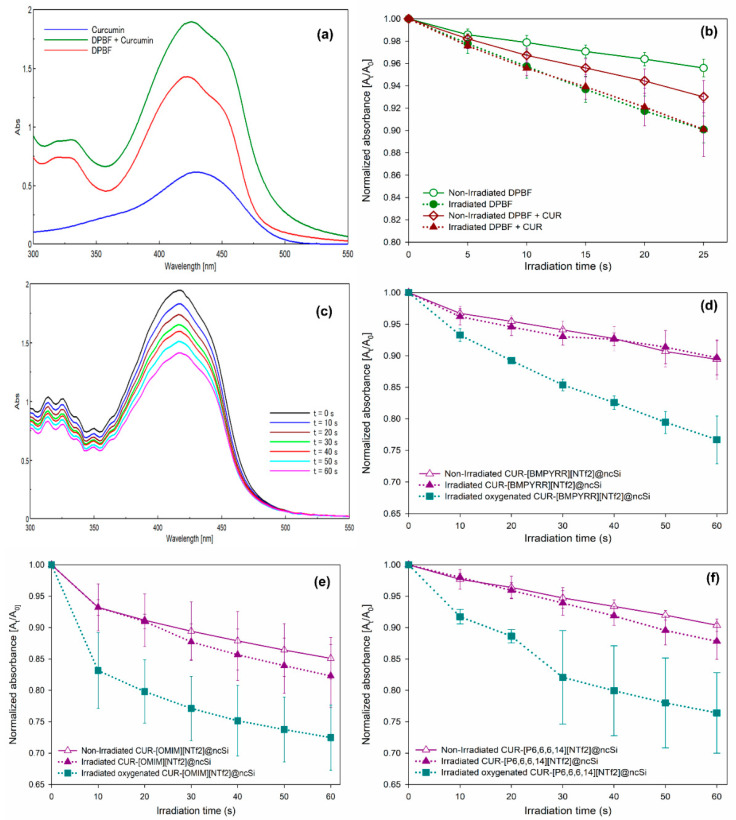
Singlet oxygen,^1^O_2,_ detection by a chemical probe (DPBF) in aerated and oxygen-saturated (oxygenated) solutions or suspensions: (**a**) UV-VIS absorbance spectrum of DPBF, curcumin, and DPBF + curcumin in EtOH:water solutions (1:9 *v*/*v*); (**b**) normalized absorbance at 416 nm of a DPBF solution (1:9 EtOH:water *v*/*v*) and DPBF+curcumin solution (1:9 EtOH:water *v*/*v*), in the dark or after successive irradiation with blue light; (**c**) changes in the absorbance spectrum of DPBF in CUR-[BMPYRR][NTf_2_]@ncSi suspensions upon successive irradiation; (**d**–**f**) normalized absorbance at 416 nm of DPBF in CUR-IL@ncSi suspensions (1:9 EtOH:water *v*/*v*): in the dark; upon successive irradiation; and upon successive irradiation following oxygenation of the suspensions.

**Figure 7 pharmaceutics-15-01080-f007:**
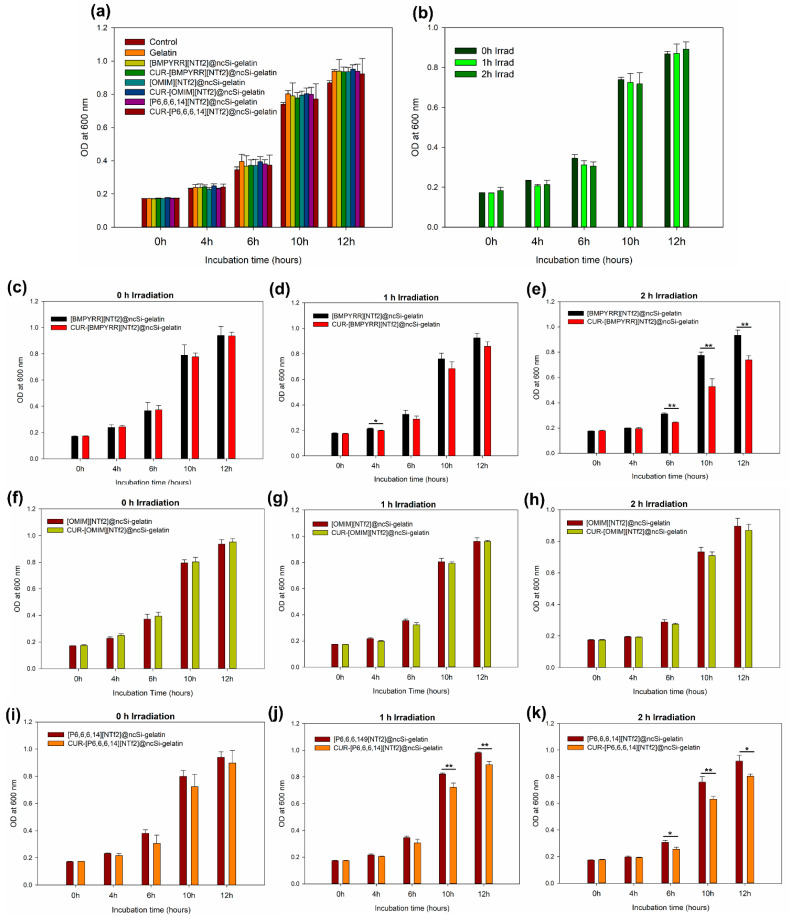
Optical density (OD) of *L. rhamnosus* cultures at different incubation times after cells were exposed to different treatments. (**a**) After 30 min of contact in the dark with the bottom of the culture well (control), gelatin film, gelatin films containing IL@ncSi, and gelatin films containing CUR-IL@ncSi; (**b**) control group after irradiation with blue light during 0 h, 1 h, or 2 h; (**c**–**e**) after contact with [BMPYRR][NTf_2_]@ncS-gelatin film or CUR-[BMPYRR][NTf_2_]@ncSi-gelatin film and irradiation with blue light for 0 h, 1 h, or 2 h; (**f**–**h**) after contact with [OMIM][NTf_2_]@ncS-gelatin film or CUR-[OMIM][NTf_2_]@ncSi-gelatin film and irradiation with blue light for 0 h, 1 h, or 2 h; (**i**–**k**) after contact with [P_6,6,6,14_][NTf_2_]@ncS-gelatin film or CUR-[P_6,6,6,14_][NTf_2_]@ncSi-gelatin film and irradiation with blue light for 0 h, 1 h, or 2 h. * *p* < 0.05; ** *p* < 0.01.

**Table 1 pharmaceutics-15-01080-t001:** Name, chemical structure, solubility in water, and oxygen solubility of the three ILs used in this work.

IL	Name	Chemical Structure	Solubility in Water (mol/mol)	O_2_ Solubility [mM]
[BMPYRR][NTf_2_]	1-Butyl-1-methylpyrrolidinium bis(trifluoromethylsulfonyl)imide	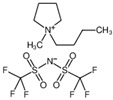	2.38 × 10^−4^ (25 °C) [38]	9.1 ± 0.9 (25 °C, 1 atm O_2_) [40]
[OMIM][NTf_2_]	1-octyl-3-methylimidazolium bis(trifluoromethylsulfonyl)imide	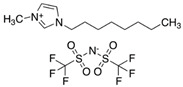	3.4 × 10^−5^ (25 °C) [37]	8.3 ± 0.8 (25 °C, 1 atm O_2_) [41]
[P_6,6,6,14_][NTf_2_]	trihexyltetradecylphosphonium bis(trifluoromethylsulfonyl)imide	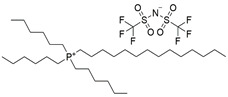	Not available	6.0 ± 0.5 (35 °C, 1 atm O_2_) [42]

**Table 2 pharmaceutics-15-01080-t002:** Amount of IL (%, *w*/*w*) incorporated into NCs determined by TGA and by EA.

NC	TGA	EA
CUR-[BMPYRR][NTf_2_]@ncSi	64.4 ± 11.4	35.7 ± 1.6
CUR-[OMIM][NTf_2_]@ncSi	94.0 ± 1.2	83.6 ± 1.5
CUR-[P_6,6,6,14_][NTf_2_]@ncSi	94.8 ± 2.2	94.4 ± 1.1

**Table 3 pharmaceutics-15-01080-t003:** ^1^O_2_ lifetimes (µs) in the different tested samples.

Sample	[BMPYRR][NTf_2_]	[OMIM][NTf_2_]	[P_6,6,6,14_][NTf_2_]
CUR-IL solution	35	27	28
CUR-IL@ncSi suspension	4.5	12	16

## Data Availability

The data presented in this study are available upon request from the corresponding author.

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
