# Peer review of "Novel Oxygen- and Curcumin-Laden Ionic Liquid@Silica Nanocapsules for Enhanced Antimicrobial Photodynamic Therapy"

_pharmaceutics, 2023, doi:10.3390/pharmaceutics15041080_

Round 1

Reviewer 1 Report

The article entitled Novel oxygen- and curcumin- laden ionic liquid@silica nanocapsules for enhanced antimicrobial photodynamic therapy is a document of interesting subject matter due the interest in new nano-carriers for drug delivery.  This manuscript presents new useful information, the results are well documented, and the experimental technique and processing of the data meet high standards.

Therefore, I think this paper is a fine contribution to the journal. However, it needs some major changes before being accepted. Make the following corrections:

1.      'Title' seems very unique as per current research trends. While, 'Abstract' should more focus on main research outcomes and novelty should mention, which is missing. Please Add 1 or 2 lines as per novelty of work for indicating importance the Novel oxygen- and curcumin- laden ionic liquid@silica nanocapsules for enhanced antimicrobial photodynamic therapy in the 'Abstract' section.

2.      'Introduction' section not enough discussed with recent updates on  nanocarriers for drugdelivery to treat cancers  should follow the cited links given:

-  https://doi.org/10.1016/j.jddst.2022.103982

- https://doi.org/10.1016/j.molliq.2022.120003

-https://doi.org/10.1016/j.jddst.2023.104285

- https://doi.org/10.1016/j.jddst.2022.103600

- https://doi.org/10.1016/j.ijbiomac.2023.123766

- https://doi.org/10.1016/j.matchemphys.2023.127336

3.      The structural formula of the drug should be given in the text.

4.      Scale bar is not clear in Fig. 2 (a-b).

5.      Why did the authors choose the ionic liquid@silica nanocapsules in the work for delivery of the drug and no other nanocarreiers like micelles, microemulsions, niosomes?

6.      Why did the authors choose the ionic liquid in the current nanocarrier and no other compounds.

7.      Please try to characterize nanocarrier size without the drug. It is a suggestion.

8.      Does the size and structure of the NP play an important role?It is noteworthy that 100 nm particles or larger generally do not penetrate well throughout the tumour mass, and smaller nanoparticles do not accumulate sufficiently in the tumour vasculature by the enhanced permeability and retention (EPR) effect and do not achieve good tumour penetration.

9.      In some sections of results, there is a lack of thorough discussion on the results. The authors should be more informative and provide more comparison between the results of the current work with former studies.

10.  How likely is the new agent approved by FDA?

11.  The stability of the systems over time should be disclosed.

12.  In the conclusion, provide a brief explanation about the future perspective of the developed carrier and how it can be modified to exhibit better performance. In the other words, The authors should do the analysis the conclusion section must clearly establish a strong correlation with the proposed topic.

13.  Your abstract should clearly state the essence of the problem you are addressing, what you did and what you found and recommend. That will help a prospective reader of the abstract to decide if they wish to read the entire article.

14.  The authors should not use undefined abbreviations (ncSi) in the abstract

15.  In the section 2.2, the amount of curcumin was not added. Please rewrite method, readers will be understood and can be not used for their Lab.

16.  Amount of Nanocapsules was mentioned in Antimicrobial photodynamic activity. Method is obscure.

17.  ‘Reveling a size distribution’ reveling is grammer wrong. These results clearly show that 387 the maximum concentrations of dissolved O2 observed. This sentence has grammer problem. Please check all MS for grammer.

18.  The morphology of IL will be compared with similar studies.

As it stands, concern author should give another chance to revise their article and should highlight in the revised manuscript text, so far recommended that the article is mandatory for 'Major revision'.

Reviewer 2 Report

Dear authors

Let me congratulate you for such interesting and accurate work. In every paragraph there is a lot of interesting information that is worthy to read, including the references used, that I find, especially useful.

The description and discussions are impeccable.

In my opinion, as it is, the paper is ready to be published

Many thanks for the work

Reviewer 3 Report

Photodynamic therapy is one of the hottest trends in personalized medicine in general and theranostics in particular.

In general, the work is well structured and logical. The literature review is relevant and shows the current state of affairs in the industry.

I cannot but note the presence of an informative ESI section.

In my opinion, there are 3 main "disadvantages".

The first is the area of excitation of the carrier and release of singlet oxygen. For the practical application of this technique, it is necessary to use the wavelength of infrared transparency of the body. At the moment, the wavelength is ~ 425 nm, which is close to ultraviolet radiation, which adversely affects a living organism.

As I understand it, setting the system to upconversion absorption, for example with rare earth elements, can solve this problem. Which is probably a development of this work.

second. Luminescence is probably the most convenient way to detect singlet oxygen, but is not accurate. I would recommend to the authors to study the kinetics of formation by the method of electron paramagnetic resonance.

Third. The issue of toxicity has not been considered. I would recommend at least hemolyzing your compounds, naturally before the release of oxygen.

In general, the work left a very positive impression, and is a complete study. In my opinion, no modifications are needed.

Round 2

Reviewer 1 Report

Accept